# Fatigue Analysis of Axisymmetric Chiral Cellular Structures Made out of 316L Stainless Steel

**DOI:** 10.3390/ma17246152

**Published:** 2024-12-16

**Authors:** Žiga Žnidarič, Branko Nečemer, Nejc Novak, Srečko Glodež

**Affiliations:** Faculty of Mechanical Engineering, University of Maribor, Smetanova 17, 2000 Maribor, Slovenia; ziga.znidaric@um.si (Ž.Ž.); branko.necemer@um.si (B.N.); n.novak@um.si (N.N.)

**Keywords:** cellular structures, chiral structures, fatigue, strain-life approach, numerical analysis

## Abstract

In the proposed study, the fatigue analysis of an axisymmetric chiral cellular structure and its modified form, made of stainless steel 316L, is carried out. The main goal of the original structure geometry was to absorb as much mechanical energy as possible with its auxetic behaviour. However, it was found through testing that its response could be improved by modifying the thickness of the struts through the structure. Representative models for the original and modified geometries were generated using a script adapted for this numerical simulation. Three different types of displacement in the shape of sine waves were used to load the structures. A hexagonal mesh was assigned and determined by convergence analysis. An existing material model with the necessary LCF parameters was assigned in the computational analyses. The data from multiple simulations were recorded and presented in graphs that showed how the fatigue life of the structures changed depending on the level of strain. We also analysed stresses and plastic deformations that occur in the structures. The results showed that, despite a better stress distribution, the fatigue life of the optimised structure was shorter in all cases.

## 1. Introduction

Cellular structures represent a unique opportunity for adoption in lightweight structures due to their favourable characteristics regarding sound insulation, damping, energy absorption, recycling, etc. [1,2,3,4,5,6]. Cellular structures may be divided into open-cell, closed-cell, and honeycomb structures [7,8]. Furthermore, we can classify them depending on whether their cells are arranged orderly or disorderly [9,10,11]. The properties of cellular structures can be engineered in advance by changing the base material, morphology, and topology of the unit cell used and its relative density. Common ways to engineer uniform cellular structures are with a wireframe based on geometric shapes and mathematical algorithms, and by changing the cell’s topology to achieve specific performance under known conditions [12,13,14]. Furthermore, cellular structures may also be used in different medical applications (i.e., cardiovascular stents [15,16,17,18]).

In recent years, different types of cellular structures have been developed to satisfy some novel properties in different engineering applications. One of them is auxetic cellular structures, which have some unique and superior mechanical properties if compared to conventional cellular structures. As a consequence of the structural deformation of internal cells during external loading, they exhibit a negative Poisson’s ratio, i.e., they significantly increase in volume if stretched and vice versa [19,20,21,22,23]. An example of cellular structures that often have an auxetic response is chiral structures [24,25,26,27]. Chirality is a phenomenon that is primarily used to describe an object whose mirror image is not identical to its original image [28]. In the case of cellular structures, the way they deform under loading is also described. Straight or curved struts are oriented in such a way that they spin around the node at which they are connected. In the medical field, they are experimenting with using these structures as bone stents. This is because of their tunable stiffness which better resembles the characteristics of real bone and avoids excess stress around the implant area. Another use is in the form of multifunctional panels, where the lattice structures serve as load-bearing members and additionally provide cooling in aircraft [29,30,31]. Another interesting implementation of lattice structures is to already existing designs. In automotive engineering, they would enable advanced design optimisation, as presented in [32], to improve crash absorption while reducing weight.

The analysed axisymmetric chiral structures were previously designed, tested, and characterised by Novak and Mauko [33,34]. The original structure was developed by taking a conventional tetra-chiral unit cell and mapping it to an axisymmetric space. The newly developed structure was later fabricated using additive manufacturing, specifically a powder bed fusion system, and tested under quasi-static and dynamic regimes. The results were validated using computer simulations. In the studies, it was concluded that these types of structures exhibit a more gradual increase in Poisson’s ratio and stiffness under dynamic loads than conventional cellular materials. Based on these results, a new structurally optimised geometry was designed to achieve a more uniform stress field and greater stiffness. This was possible by altering strut diameters in specific areas.

Several researchers have recently investigated the fatigue behaviour of auxetic structures. Nečemer et al. [35] investigated the fatigue characteristics of re-entrant auxetic structures, focusing on the orientation of their unit cells. Their results confirmed that fatigue cracks typically initiated at points of highest stress concentration, often at the corners of the base cell, and propagated until the structure ultimately failed. Glodež et al. [36] and Tomažinčič et al. [37] explored the Low Cycle Fatigue (LCF) behaviour of auxetic structures made from high-strength aluminium alloy 7075-T651, taking surface roughness into account. They found that surface roughness significantly impacted the structure’s fatigue life, especially under small amplitude strains within the High Cycle Fatigue (HCF) range. Furthermore, Peta et al. [29] concluded that the surface roughness of aluminium alloys would influence functional surface features such as wetting and lubrication, which could play a role in a structure performance under cyclic loading. Nečemer et al. [38] studied the fatigue performance of two-dimensional auxetic structures made of aluminium alloy 5083-H111 using an inelastic energy-based approach. In their study, the higher fatigue strength of the chiral auxetic structure was obtained and compared to the fatigue strength of the re-entrant auxetic structure. Their findings revealed that fatigue fractures occurred at the intercellular links rather than at the expected nodal notched areas. In a subsequent study [39], the same authors extended their research with statistical analysis, presenting the fatigue resistance of both chiral and re-entrant auxetic structures in three distinct formats. This work demonstrated the potential to analyse the fatigue behaviour of auxetic structures while accounting for variations in stiffness. Ulbin et al. [40] investigated the fatigue behaviour of five optimised auxetic cellular structures made from AlSi10Mg aluminium alloy. They used the strain-life approach to determine the fatigue life, considering the influence of the radius of the intercellular connections. Their results indicated that the radius significantly affected the fatigue life, primarily due to the stress concentration at these points. Kolken et al. [41] studied the fatigue performance of auxetic meta-biomaterials, explicitly focusing on their High Cycle Fatigue (HCF) behaviour. The additive-manufactured (AM) auxetic meta-biomaterials, made from commercially pure titanium, demonstrated suitable morphological and mechanical properties for use in bone implantations. Tomažinčič et al. [42] investigated the static and Ultra-Low Cycle Fatigue (ULCF) behaviour of planar cellular structures made from aluminium alloy 7075-T651, employing both experimental and computational methods. Their computational analyses utilised an energy-based Continuum Damage Mechanics (CDM) approach with an explicit dynamic solver. The results showed that fatigue fractures occurred at the intercellular links rather than at the expected nodal notched geometries. The authors concluded that energy-based methods were effective for predicting the durability of cellular structures and enhancing the understanding of their fatigue performance.

This study is a continuation of the author’s previous work [27,33], in which novel 3D axisymmetric chiral auxetic structures were analysed with respect to the energy absorption capability. It was found that the regular axisymmetric chiral structure (Figure 1a) exhibits a nonuniform stress distribution through the geometry in its initial configuration, which is also reflected in its lower energy absorption capabilities. For that reason, a new structure with spatially graded porosity was developed (Figure 1b) using the structural optimisation technique. It was found that a new stiffer structure provides approximately 4.25 times higher energy absorption capability than a regular structure. However, the fatigue resistance of the analysed structures was still open and subsequent fatigue analyses, using the strain-life approach, were needed to solve this problem. The method was chosen based on the need to predict fatigue life under cycling loading. It is worth mentioning that if we wanted to build an understanding of crack propagation and the onset of failure in our structures, fracture mechanics could be a viable option in the future, as demonstrated by Barchiesi et al. [43]. Their work analyses brittle fracture propagation in strain gradient materials using the FEniCS library, giving insights into crack growth and failure mechanisms. Because our cellular materials are characterised by curved fibres, it would be possible to use continuum models, such as in work [44], to describe them. However, using this approach, the local deformation and development of strain concentration zones will not be observed in such detail. Therefore, the comprehensive computational fatigue analysis of both structures made of stainless steel 316L was performed in this study, using discrete FE models to offer us a visual comparison with experimental results. The computational analyses have been performed in the framework of 7 and [45]. Three different loading types were considered in the form of displacement given in advance, and the fatigue life of structures was then obtained using the strain-life approach.

## 2. Materials and Methods

### 2.1. Geometry and Material

The geometries that were used in computational simulations (see Figure 1) were created with the help of a Python script in Ansys Mechanical 2024 R1 and with the same parameters as described in the author’s previous work [27,33]. As we did not use the entire structure in numerical simulations, the code was modified to generate only one layer of unit cells and struts, as shown in Figure 2. Using the linear pattern and mirror command, a structure was generated that was eleven unit cells tall and three struts wide. These were then assigned a strut thickness of 0.34 mm for the regular structure and ranging from 0.10 to 0.90 mm for the graded structure. Lastly, four planes were created. Two were used to cut the models vertically, leaving us with two struts or 1/14th of the entire structure. The remaining planes cut the first and last cells horizontally. The resulting geometries are shown in Figure 3, with the regular geometry being 20.968 mm tall and the graded geometry 21.498 mm tall.

The material of both structures used in this study was stainless steel 316L (yield stress *R*_e_ = 380 MPa, Young’s modulus *E* = 194,323 MPa). The required material parameters for the determination of the stress/strain field in analysed structures and, consequently, to determine the expected fatigue life were taken from [46]. The parameters used for modelling material elastic and plastic behaviour used the Chaboche kinematic hardening model. Here, *C*_1_ = 320,000 MPa and γ_1_ = 5500 describe the initial hardening when the material starts to yield. Furthermore, *C*_2_ = 97,000 MPa and γ_2_ = 1000 model the transition at medium strain, and the last two constants, *C*_3_ = 25,000 MPa and γ_3_ = 150, describe the slope at high strain amplitude. Furthermore, the following Low Cycle Fatigue parameters have been considered when determining fatigue life: *σ_f’_* = 1018.6 MPa, *b* = −0.085, *ε_f’_*= 0.668, and *c* = −0.595.

### 2.2. Computational Analyses

The next step in the computational analysis was defining the boundary conditions. Frictionless supports were assigned to the vertical faces of both models (Figure 4a). These restrict perpendicular movement while allowing all other forms of translation. Seven of the eight bottom surfaces were assigned the same boundary condition. The remaining surface nearest to the centre was fixed in place with no degree of freedom to prevent any instabilities in the simulation (Figure 4c). Lastly, a sine wave displacement was applied to all horizontal remaining faces to simulate an external load (Figure 4b). Three types of loading (*R* = −1, *R* = 0 in compression, *R* = 0 in tension) were used, as shown in Figure 5. For each type of loading, six load cases were considered, resulting in the total strain range Δ*ε* of the whole structure ranging from 0.9% to 1.4% of its total height. All displacements were assigned in the vertical axis (z-axis in Figure 3). This is because boundary conditions and geometry were designed for vertical loads, ensuring consistency with previous studies. In addition, we wanted to avoid inaccuracies and inconsistencies that might occur from applying complex forces to structures optimised for vertical loading.

After all the boundary conditions were assigned, the meshing of the two models followed. Convergence analysis was carried out on a smaller model of the original geometry to determine the optimal size of finite elements. Instead of displacement, a force of 35 N was applied to the top of the structure, and maximum displacements were tracked (see Figure 6). A uniform element size of 0.1 mm was used for the original geometry. Because of the different sizes of struts, this mesh size was unsuitable for the entire optimised geometry. The smallest two strut diameters were meshed with a size of 0.03 mm and 0.04 mm, respectively, while the struts with the largest diameter used an element size of 0.15 mm to reduce the size of the numerical model. Throughout these changes, we ensured that the number of elements across the strut diameter was equal to or greater than the number of elements in the original structure. The elements were hexagonal with quadratic interpolation for both models.

In the subsequent computational analyses, the fatigue life of the analysed structures was obtained using the strain-life approach and with consideration of a Morrow mean stress correction using the following equation:(1)$$\varepsilon_{a}=\frac{∆\varepsilon }{2}=\frac{∆\varepsilon_{e}}{2}+\frac{∆\varepsilon_{p}}{2}=\frac{\sigma_{f}^{\prime }-\sigma_{m}}{E}\cdot {\left(2\cdot N_{i}\right)}^{b}+\varepsilon_{f}^{\prime }\cdot {\left(2\cdot N_{i}\right)}^{c}$$

In Equation (1), *ε_a_* is the local amplitude strain, *E* is the Young’s modulus, *σ_f’_* is the fatigue strength coefficient, *b* is the fatigue strength exponent, *ε_f’_* is the fatigue ductility coefficient, *c* is the fatigue ductility exponent, and *σ_m_* is the mean stress. Here, it should be pointed out that the magnitude of *N* in Equation (1) represents the number of loading cycles for crack initiation inside the cell strut of the analysed cellular structure. The strain-life approach was chosen to account for both elastic and plastic deformation during cyclic loading.

## 3. Results and Discussion

Figure 7 shows the equivalent von Mises stress distribution in both analysed structures for the load ratio *R* = −1 and strain range of the whole structure Δ*ε* = 1.4%. The maximum stresses occur on the vertical struts. In the original structure (Figure 7a), the equivalent stress is distributed nonuniformly across the individual cells (the inner cells are exposed to higher stresses than the outer cells). On the other hand, the optimised structure (Figure 7b) demonstrates a more uniform equivalent stress distribution on the inner and outer cells. The values of the maximum equivalent stresses are quite similar for both analysed structures (690.3 MPa for the original structure and 697.4 MPa for the optimised structure). It is evident that the maximum equivalent stress exceeds the yield stress of the material (*R*_e_ = 380 MPa; see Section 2.1) significantly, which leads to the local plastification and, consequently, to the initiation of initial fatigue cracks in those locations. Similar calculations may also be performed for other types of loadings (*R* = 0 in tension and *R* = 0 in compression) and other strain ranges of the whole structure (Δ*ε* = 0.9% to 1.4%). The complete computational results are presented in Table 1.

Figure 8 shows a side view of both structures in their deformed and undeformed states. We can observe that under a compressive load, the entire model wants to shift to the right, which would be the centre when looking at the whole structure (see also Figure 1). This shows confirmation that both geometries have an auxetic effect as documented in our previous studies.

Figure 9 shows the local equivalent plastic strain distribution in both analysed structures for the load ratio *R* = 0 in tension and strain range of the whole structure Δ*ε* = 1.4%. The maximum equivalent plastic strains occur at the same locations as maximum equivalent stresses (see Figure 7). Their values are 3.8% for the original structure and 4.7% for the optimised structure. Similar calculations may also be performed for other types of loadings (*R* = −1 and *R* = 0 in compression) and other strain ranges of the whole structure (Δ*ε* = 0.9% to 1.4%). The complete computational results are presented in Table 2.

Figure 10 shows the fatigue life (i.e., the number of loading cycles *N*) for both analysed structures in dependence on the strain range of the whole structure Δ*ε* and type of loading. It is clear that for all types of loading (*R* = −1, *R* = 0 in compression, *R* = 0 in tension), the power trend lines for the optimised geometry had a steeper slope and lie to the left of the lines of the original structures. This means that, at the same strain range of the whole structure Δ*ε*, the optimised structure (i.e., the structure with graded porosity) is less resistant to fatigue loading if compared to the original geometry. This could be explained by the fact that the optimised structure is stiffer, leading to higher stresses and local plastic strains in critical locations of the structure. Furthermore, a smaller radius between the individual cells appears in the optimised structure due to graded porosity (the thickness of the struts increases in the radial direction from the inner to the outer side of the structure), leading to higher stress concentrations and, consequently, shorter fatigue life. The trend lines and data show a strong correlation evidenced by the high *R^2^* values. They were calculated by comparing the squared differences between the actual data points and the predicted values from the trend lines, with higher values indicating a better fit.

## 4. Conclusions

In this study, the computational fatigue analysis of an axisymmetric chiral cellular structure (original structure) and its modified form (optimised structure) made of stainless steel 316L was carried out in the framework of Ansys software. The study is a continuation of the author’s previous work, where novel 3D axisymmetric chiral auxetic structures were analysed with respect to the energy absorption capability. However, the fatigue resistance of the analysed structures was still open, and subsequent fatigue analyses were needed to solve this problem. Therefore, the comprehensive computational fatigue analysis of both structures was performed in this study. Based on the obtained computational results and their evaluation, the following conclusions can be made:The original axisymmetric chiral structure exhibits a nonuniform stress distribution, which is reflected in its lower energy absorption capabilities. However, a new optimised structure with spatially graded porosity was found to be stiffer and had significantly higher energy absorption capability.The maximum equivalent plastic strains occurred at the same locations as maximum equivalent stresses. Their values for the optimised structure were higher than those for the original structure.The optimised structure (i.e., the structure with graded porosity) was found to be less resistant to fatigue loading when compared to the original geometry. This could be explained by the fact that the optimised structure is stiffer, which consequently leads to higher stresses and local plastic strains in critical locations of the structure. Furthermore, the optimised structure demonstrates a smaller radius between individual cells due to graded porosity (i.e., different thicknesses of cell struts), which represents a higher notch effect and, consequently, shorter fatigue life.Experimental fatigue testing on both original and optimised structures should be performed in further research to confirm the computational results.

The modifications made to the axisymmetric chiral cellular structure gave it better energy absorption and stress distribution when analysed with static and impact tests. However, they shortened their fatigue life when put under cyclic loads. In future, some additional optimisation strategies could be implemented. The first would be the modification of sine wave shapes used to build the structures. In our case, both stemmed from the same spline model. The next would be smoothing the areas with higher plastic strains and notch effects as they are the ones observed to have the shortest fatigue life. Lastly, introducing some additional stress redistribution features at intercellular connections might improve their performance without compromising their structural efficiency.

## Figures and Tables

**Figure 1 materials-17-06152-f001:**
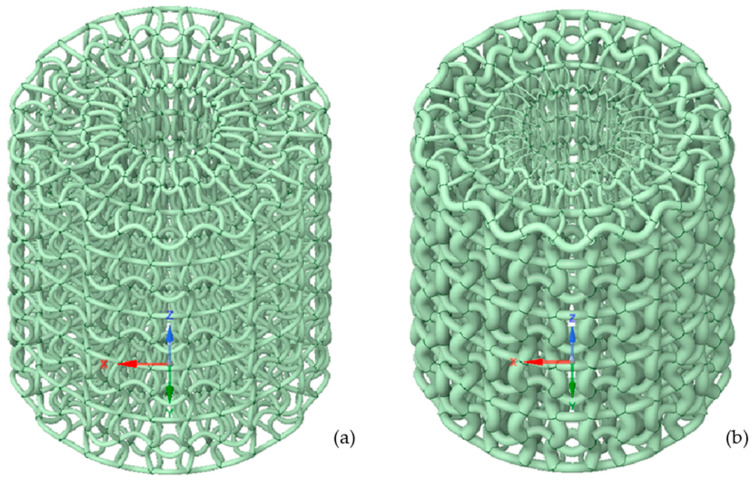
Axisymmetric chiral cellular structures: (**a**) regular structure (original), (**b**) structure with graded porosity (optimised).

**Figure 2 materials-17-06152-f002:**
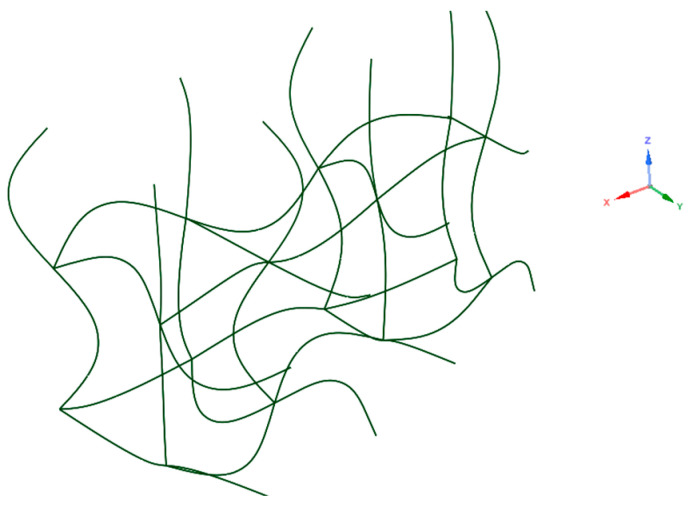
One layer of the structure modelled with splines.

**Figure 3 materials-17-06152-f003:**
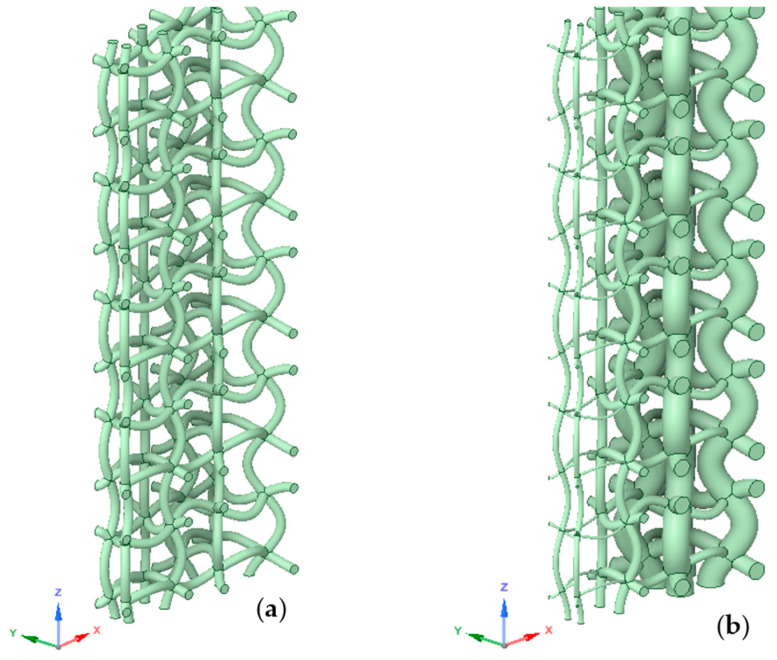
Computational model of analysed structures: (**a**) regular structure (original), (**b**) structure with graded porosity (optimised).

**Figure 4 materials-17-06152-f004:**
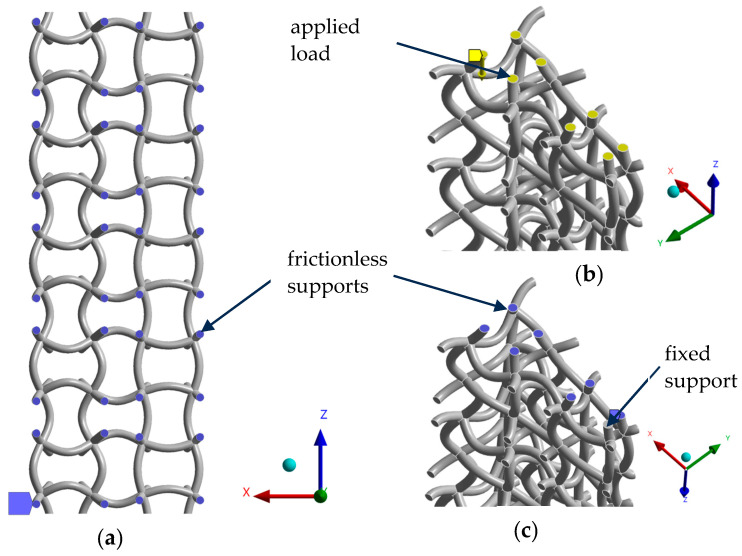
Boundary conditions: (**a**) frictionless supports on vertical faces, (**b**) applied load on horizontal faces, (**c**) frictionless supports on horizontal faces.

**Figure 5 materials-17-06152-f005:**
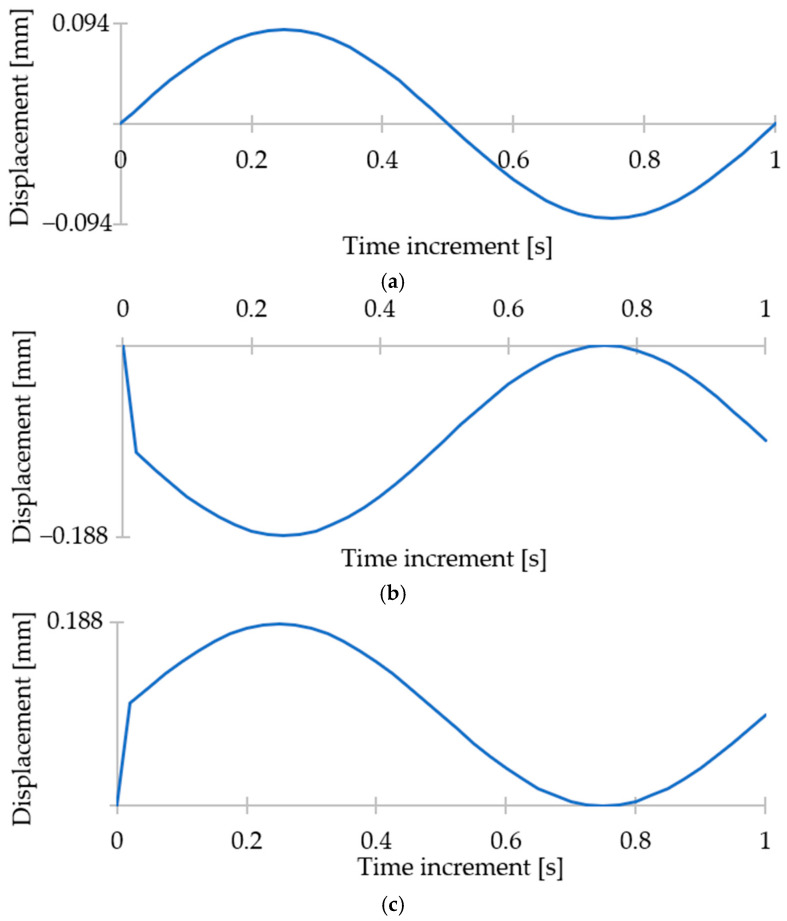
Load cases 1 (Δ*ε* = 0.9%): *(***a**) *R* = −1, (**b**) *R* = 0 in compression, (**c**) *R* = 0 in tension.

**Figure 6 materials-17-06152-f006:**
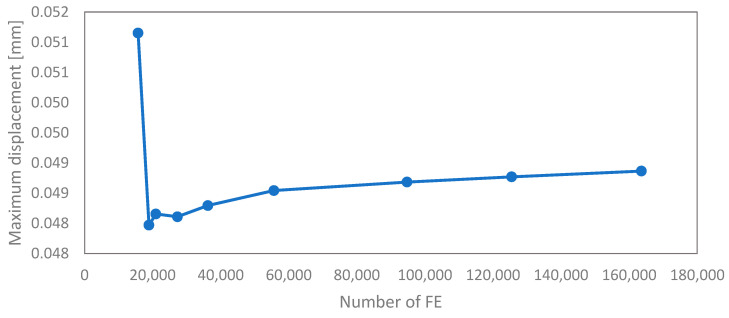
Convergence analyses of the original structure.

**Figure 7 materials-17-06152-f007:**
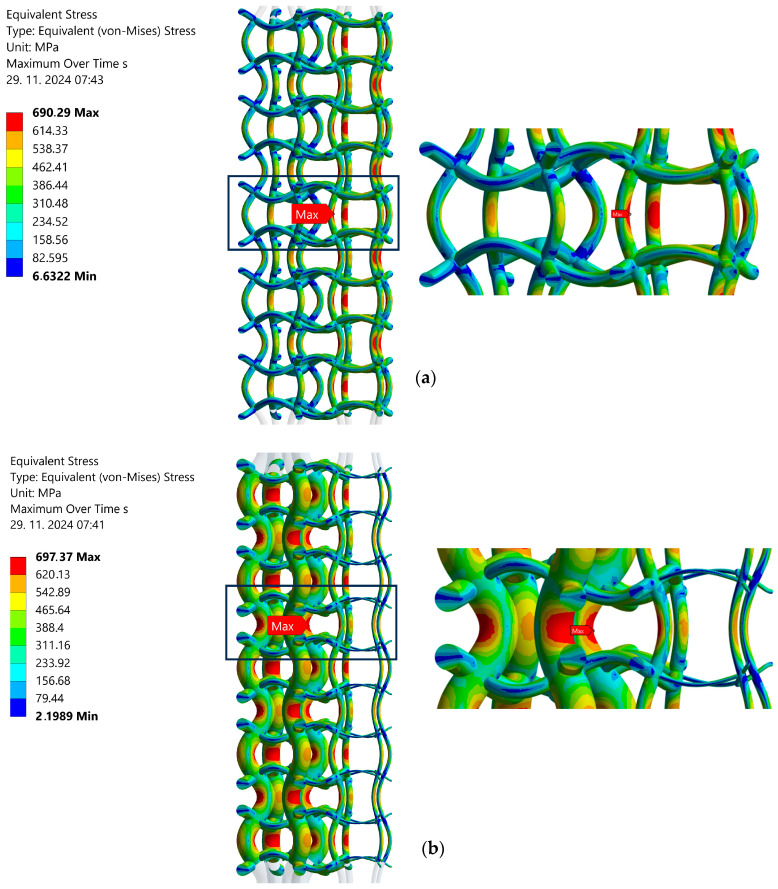
Mises equivalent stress distribution (*R* = −1, Δ*ε* = 1.4%): (**a**) regular (original) structure, (**b**) structure with graded porosity (optimised).

**Figure 8 materials-17-06152-f008:**
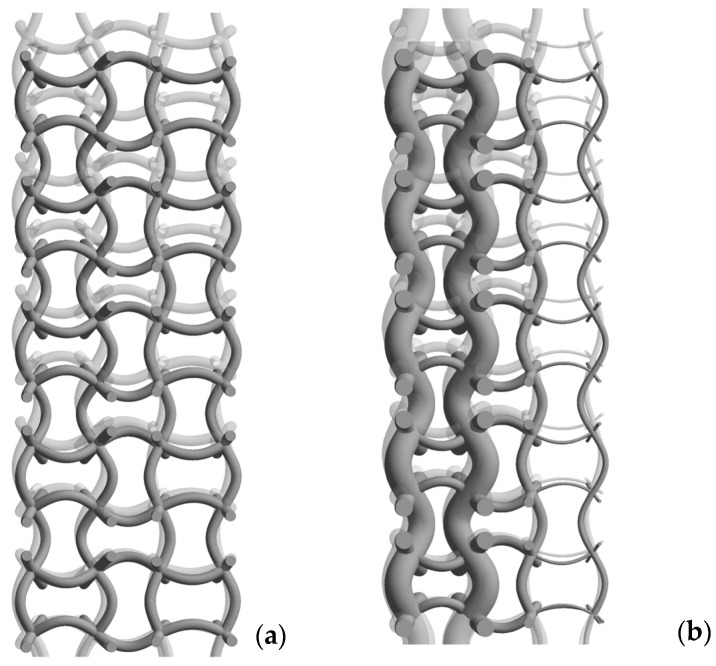
Auxetic effect of analysed structures: (**a**) regular (original) structure, (**b**) structure with graded porosity (optimised).

**Figure 9 materials-17-06152-f009:**
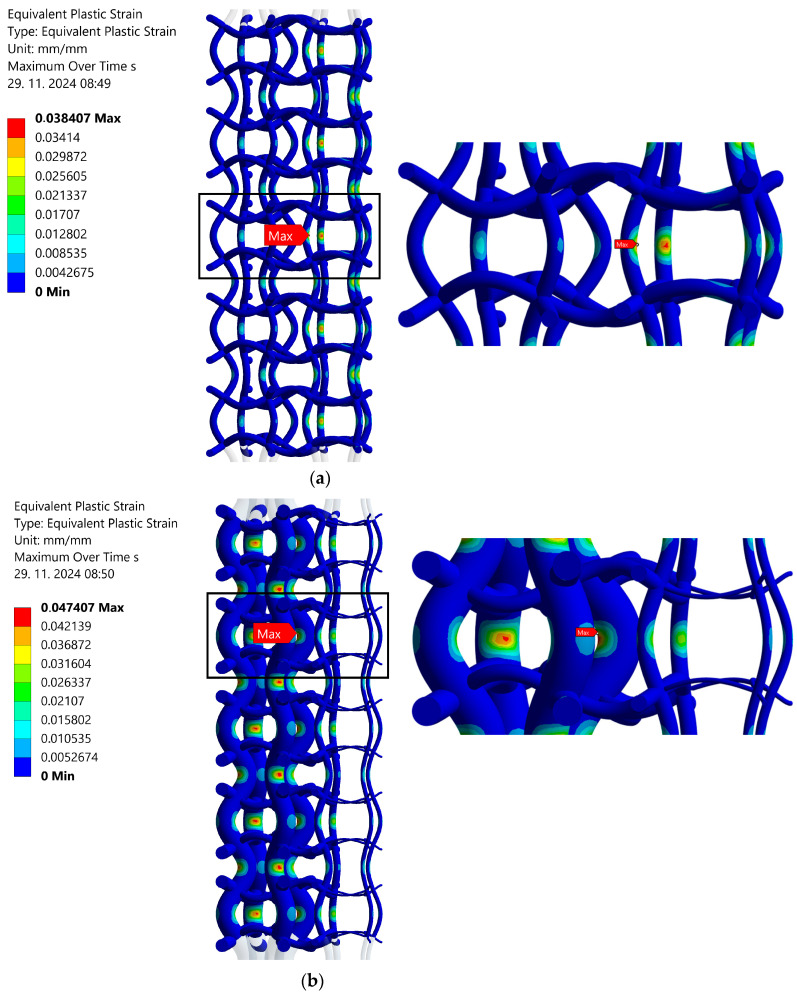
Local equivalent plastic strain (*R* = 0 in tension, Δ*ε* = 1.4%): (**a**) regular structure (original), (**b**) structure with graded porosity (optimised).

**Figure 10 materials-17-06152-f010:**
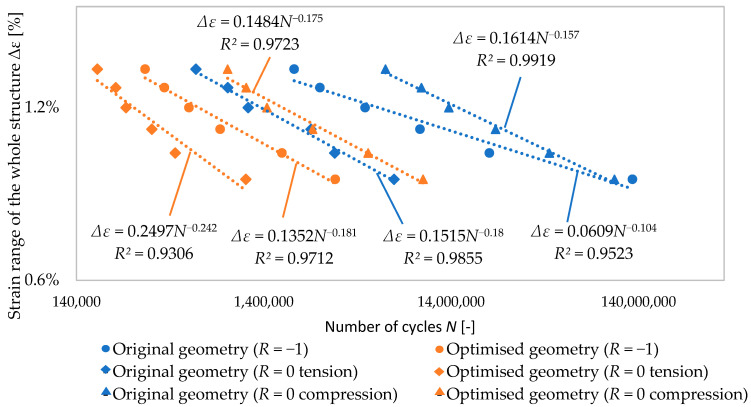
Fatigue life of analysed structures.

**Table 1 materials-17-06152-t001:** Maximum Mises equivalent stress in the analysed structures.

Strain Range of the Whole Structure Δ*ε* [%]	σ_eq_^max^ [MPa]					
	Original Structure			Optimised Structure		
	*R* = −1	*R* = 0 Compression	*R* = 0 Tension	*R* = −1	*R* = 0 Compression	*R* = 0 Tension
0.9	633.35	689.74	690.33	675.66	695.21	695.13
1.0	654.74	694.51	694.99	684.10	697.15	697.31
1.1	667.96	697.48	697.86	690.84	698.51	698.66
1.2	678.17	699.26	699.56	694.03	699.39	699.65
1.3	685.72	700.30	700.73	696.00	699.99	700.05
1.4	690.29	700.99	701.20	697.37	700.26	700.61

**Table 2 materials-17-06152-t002:** Maximum local equivalent plastic strain in the analysed structures.

Strain Range of the Whole Structure Δ*ε* [%]	ε_eq_^max^ [%]					
	Original Structure			Optimised Structure		
	*R* = −1	*R* = 0 Compression	*R* = 0 Tension	*R* = −1	*R* = 0 Compression	*R* = 0 Tension
0.9	0.45	1.76	1.79	0.86	2.56	2.52
1.0	0.58	2.10	2.14	1.02	2.96	2.98
1.1	0.71	2.47	2.51	1.24	3.29	3.35
1.2	0.84	2.85	2.90	1.42	3.81	3.89
1.3	0.99	3.24	3.41	1.61	4.25	4.29
1.4	1.10	3.76	3.84	1.81	4.71	4.74

## Data Availability

The original contributions presented in this study are included in the article. Further inquiries can be directed to the corresponding author.

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
