# Peer review of "Fatigue Analysis of Axisymmetric Chiral Cellular Structures Made out of 316L Stainless Steel"

_materials, 2024, doi:10.3390/ma17246152_

Round 1
Reviewer 1 Report
Comments and Suggestions for Authors
Dear Authors,
the paper is interesting and generally well written. The aim of the work includes the analysis of the original geometry of the structure, which could absorb as much mechanical energy as possible is an important issue from a practical and scientific point of view. Therefore, the subject of the work and the presented research results are useful.
However, there are a few issues that require more detailed explanation.
1. Line 10 - "original geometry" - requires adding what exactly is to have the original geometry. I assume that adding the word "structure" will be sufficient - original structure geometry.
2. Line 31 - give ways in which these structures can be engineered.
3. Lines 53-55 - It is true that the fatigue life of the structure depends on the surface roughness. It is worth adding one more citation here: https://doi.org/10.3390/ma17235716 This paper also states that the surface roughness of aluminum alloys influences a number of functional surface features. Thus, it affects fatigue life, but also lubrication and wetting in tribological systems.
4. Line 94 - please provide version of Ansys software, and tools used.
5. Fig 6, 7 - would be good to prepare new images with exactly the same scale. Then two models could be compared.
6. Please describe Figure 8 in more detail. How the coefficient of determination R2 was calculated. What type of trend line was used.
7. What is the mass of the material in the original structure and in the optimized one? Could the optimized structure be more difficult to implement from a manufacturing perspective? Strength properties are one point of view, but how does this translate into economic aspects?
Author Response
The paper is interesting and generally well written. The aim of the work includes the analysis of the original geometry of the structure, which could absorb as much mechanical energy as possible is an important issue from a practical and scientific point of view. Therefore, the subject of the work and the presented research results are useful. However, there are a few issues that require more detailed explanation.
Comment #1: Line 10 - "original geometry" - requires adding what exactly is to have the original geometry. I assume that adding the word "structure" will be sufficient - original structure geometry.
Response: As suggested, the word “structure” was added in line 10.
Comment #2: Line 31 - give ways in which these structures can be engineered
Response: The following paragraph has been added in the Introduction of the revised version of the manuscript: “Common ways to engineer uniform cellular structures are with a wireframe based on geometric shapes and mathematical algorithms and by changing the cell's topology to achieve specific performance under known conditions”.
Comment #3: Lines 53-55 - It is true that the fatigue life of the structure depends on the surface roughness. It is worth adding one more citation here: https://doi.org/10.3390/ma17235716 This paper also states that the surface roughness of aluminum alloys influences a number of functional surface features. Thus, it affects fatigue life, but also lubrication and wetting in tribological systems.
Response: The suggested reference and its findings have been added to the revised version of the manuscript (see reference [29] in the revised manuscript).
Comment #4: please provide version of Ansys software, and tools used.
Response: In the revised version of the manuscript, the Ansys software is cited as follows: Ansys Mechanical 2024 R1 [45]
Comment #5: Fig 6, 7 - would be good to prepare new images with exactly the same scale. Then two models could be compared.
Response: Figs. 6 and 7 (Figs. 7 and 9 in the revised manuscript) have been improved as suggested.
Comment #6: Please describe Figure 8 in more detail. How the coefficient of determination R2 was calculated. What type of trend line was used
Response: Fig. 8 (Fig. 10 in the revised manuscript) has been described in more detail, also considering the description of how the coefficient of determination was calculated.
Comment #7: What is the mass of the material in the original structure and in the optimised one? Could the optimised structure be more difficult to implement from a manufacturing perspective? Strength properties are one point of view, but how does this translate into economic aspects?
Response: The optimised structure was not fabricated in our previous works, so we do not have the information about its physical mass. The original structure was fabricated with additive manufacturing, and the mass is known, but it would differ from the numerical model since we added geometry to be able to apply our boundary conditions.
Reviewer 2 Report
Comments and Suggestions for Authors
Presented problem is practically and scientifically important. The searching of the auxetic structures has great meaning in further practical applications. The manuscript is well organized and written so I recommend publishing. There are some suggestions for Authors:
1. The new, scientific elements to earlier investigations should be stronger underlined in introduction.
2. The choice of the loading models should be explained.
3. The FEM model should be presented more exactly.
4. It would be very interesting if the auxetic phenomenon of presented structure would be shown graphically.
5. It is the question about structure building? Due to obtained results, which are not fully satisfying, will the new structure composite be trying to find in the further investigations?
Author Response
Presented problem is practically and scientifically important. The searching of the auxetic structures has great meaning in further practical applications. The manuscript is well organised and written so I recommend publishing. There are some suggestions for Authors:
Comment #1: The new, scientific elements to earlier investigations should be stronger underlined in introduction.
Response: In the revised manuscript, the section Introduction has been extended to describe our earlier investigations in more detail.
Comment #2: The choice of the loading models should be explained.
Response: In section 2.2 of the revised manuscript, the loading models are additionally explained as suggested.
Comment #3: The FEM model should be presented more exactly.
Response: In section 2.2 of the revised manuscript, the FEM model is presented more exactly as suggested. Furthermore, a new figure (Fig. 4) has been added for that purpose.
Comment #4: It would be very interesting if the auxetic phenomenon of presented structure would be shown graphically.
Response: A new figure (Fig. 8) has been added to the revised manuscript describing undeformed and deformed geometries to show the auxetic effect of analysed structures.
Comment #5: It is the question about structure building? Due to obtained results, which are not fully satisfying, will the new structure composite be trying to find in the further investigations.
Response: In the Conclusions of the revised manuscript, some possible optimisation strategies are proposed to improve the current computational model.
Reviewer 3 Report
Comments and Suggestions for Authors
The paper investigates the fatigue life of axisymmetric chiral cellular structures made of 316L stainless steel using computational simulations. The analysis compares a regular (original) structure and a modified (optimized) structure with graded porosity. While the optimized structure achieves better stress distribution and higher energy absorption, it demonstrates a shorter fatigue life due to higher stress concentrations and localized plastic strains. The study employs the strain-life approach for fatigue analysis and highlights the need for experimental validation of the computational findings.
The investigation of cellular structures with chiral auxetic designs is timely and aligns with emerging interests in lightweight, high-performance materials.
The side-by-side analysis of original and optimized structures is well-documented, facilitating an understanding of the trade-offs between stiffness and fatigue performance.
The paper covers stress, plastic strain distributions, and fatigue life across various loading conditions and strain ranges.
While the graded porosity approach is well-executed, similar strategies have been previously explored in cellular structure research. The innovation could be better emphasized.
The paper mentions stress concentrations due to smaller radii but does not explore potential design modifications to mitigate these effects.
Minor typographical inconsistencies, such as figure labels and references, slightly detract from readability (e.g., Figures 2 and 3 lack detailed captions).
Suggestions for Improvement:
1. Explore additional optimization strategies that might mitigate fatigue issues, such as smoothing inter-cell connections or incorporating variable radii. A comment on future development would be appreciated.
2. Expand a little more the discussion on the implications of this research for industrial applications (e.g., aerospace, biomechanics).
3. Improve figures with clearer labels and higher resolution. For example, provide zoomed-in views of stress and strain concentration zones.
4. I suggest giving more details about the model of damage/fatigue and providing some examples from the literature in regard (see, e.g., [1]).
[1] Barchiesi, E., Yang, H., Tran, C. A., Placidi, L., & Müller, W. H. (2021). Computation of brittle fracture propagation in strain gradient materials by the FEniCS library. Mathematics and Mechanics of Solids, 26(3), 325-340.
5. These cellular materials characterized by curved fibers can be described by continuum models ad hoc conceived. Is it possible to treat the proposed variants of the materials in this framework? Like, for example, done in [2]. Please provide a comment on this matter.
[2] Ciallella, A., D'Annibale, F., Del Vescovo, D., & Giorgio, I. (2023). Deformation patterns in a second-gradient lattice annular plate composed of “spira mirabilis” fibers. Continuum Mechanics and Thermodynamics, 35(4), 1561-1580.
I will suggest some revisions. The paper addresses an important topic with adequate, rigorous computational methods. However, it requires some improvements. After addressing the pointed-out issues, the paper could be published.
Author Response
The paper investigates the fatigue life of axisymmetric chiral cellular structures made of 316L stainless steel using computational simulations. The analysis compares a regular (original) structure and a modified (optimised) structure with graded porosity. While the optimised structure achieves better stress distribution and higher energy absorption, it demonstrates a shorter fatigue life due to higher stress concentrations and localised plastic strains. The study employs the strain-life approach for fatigue analysis and highlights the need for experimental validation of the computational findings.
Suggestions for Improvement:
Comment #1: Explore additional optimisation strategies that might mitigate fatigue issues, such as smoothing inter-cell connections or incorporating variable radii. A comment on future development would be appreciated.
Response: In the Conclusions of the revised manuscript, some possible optimisation strategies are proposed to improve the current computational model.
Comment #2: Expand a little more the discussion on the implications of this research for industrial applications (e.g., aerospace, biomechanics).
Response: In the revised manuscript, the section Introduction has been extended to describe some common implications of cellular structures in different aplication fields.
Comment #3: Improve figures with clearer labels and higher resolution. For example, provide zoomed-in views of stress and strain concentration zones.
Response: Some figures have been improved as suggested.
Comment #4: I suggest giving more details about the model of damage/fatigue and providing some examples from the literature in regard (see, e.g., [1]).
[1] Barchiesi, E., Yang, H., Tran, C. A., Placidi, L., & Müller, W. H. (2021). Computation of brittle fracture propagation in strain gradient materials by the FEniCS library. Mathematics and Mechanics
of Solids, 26(3), 325-340.
Response: The suggested reference and its findings have been added to the revised version of the manuscript (see reference [43] in the revised manuscript).
Comment #5: These cellular materials characterised by curved fibers can be described by continuum models ad hoc conceived. Is it possible to treat the proposed variants of the materials in this framework? Like, for example, done in [2]. Please provide a comment on this matter.
[2] Ciallella, A., D'Annibale, F., Del Vescovo, D., & Giorgio, I. (2023). Deformation patterns in a second-gradient lattice annular plate composed of “spira mirabilis” fibers. Continuum Mechanics and Thermodynamics, 35(4), 1561-1580.
Response: The suggested reference and its findings have been added to the revised version of the manuscript (see reference [44] in the revised manuscript).